# Assessing the Relationship between District and State Policies and School Nutrition Promotion-Related Practices in the United States

**DOI:** 10.3390/nu12082356

**Published:** 2020-08-07

**Authors:** Gabriella M. McLoughlin, Lindsey Turner, Julien Leider, Elizabeth Piekarz-Porter, Jamie F. Chriqui

**Affiliations:** 1Prevention Research Center, Brown School, Washington University in St. Louis, St. Louis, MO 63130, USA; 2Siteman Cancer Center and Division of Public Health Sciences, Department of Surgery, Washington University School of Medicine, Washington University in St. Louis, St. Louis, MO 63110, USA; 3College of Education, Boise State University, Boise, ID 83725, USA; lindseyturner1@boisestate.edu; 4Institute for Health Research and Policy, University of Illinois Chicago, Chicago, IL 60608, USA; jleide2@uic.edu (J.L.); epiekarz@uic.edu (E.P.-P.); jchriqui@uic.edu (J.F.C.); 5Division of Health Policy and Administration, School of Public Health, University of Illinois Chicago, Chicago, IL 60612, USA

**Keywords:** school, nutrition, legal epidemiology, school meals

## Abstract

School environments are an optimal setting to promote healthy student diets, yet it is unclear what role state and district policies play in shaping school contexts. This study examined how state and district policies are associated with school-reported practices for promoting student participation in school lunch programs. School nutrition manager data were obtained from the School Nutrition and Meal Cost Study’s (SNMCS) sample of 1210 schools in 46 states and the District of Columbia (DC) during school year 2014–2015. Relevant state laws and district policies were compiled and coded. Multivariable logistic and Poisson regressions, controlling for school characteristics, examined the relationship between state/district laws/policies and school practices. Compared to schools in districts or states with no policies/laws, respectively, schools were more likely to provide nutritional information on school meals (AOR = 2.59, 95% CI = 1.33, 5.05) in districts with strong policies, and to promote school meals at school events (AOR = 1.93, CI = 1.07, 3.46) in states with strong laws. Schools in states with any laws related to strategies to increase participation in school meals were more likely to seek student involvement in menu planning (AOR = 2.02, CI = 1.24, 3.31) and vegetable offerings (AOR = 2.00, CI = 1.23, 3.24). The findings support the association of laws/policies with school practices.

## 1. Introduction

Strong policies governing the school environment are key influencers of the implementation of evidence-based practices in schools, and policies can facilitate the development of settings that support health-promoting behaviors among students [1,2,3,4]. Numerous policies have been developed at the federal, state, district and school level, addressing practices such as nutrition education [5,6,7,8,9,10], school lunch timing [8,9,11,12] and lunch duration [8,9,13]. Such practices in schools are subsequently associated with changes in important student level outcomes, such as knowledge, behavior (consumption of fruits and vegetables) and health outcomes. Thus, it is important to understand the relationship between policies, practices and subsequent health behavior to drive policy change.

To address increasing childhood obesity rates within the United States, school administrators, teachers and other school staff have been encouraged to develop school environments that foster healthy eating and physical activity [13,14]. In 2004, the federal Child Nutrition and Women, Infants, and Children (WIC) Reauthorization Act (CNR Act) required that local education agencies (districts) participating in the federal Child Nutrition Programs (including the National School Lunch Program) adopt and implement a school wellness policy as of the 2006–2007 school year [10]. Wellness policies were required to include goals for nutrition education, physical activity and other school-based activites in order to promote student wellness. In 2010, the Healthy, Hunger Free Kids Act required that the U.S. Department of Agriculture (USDA) enhance nutrition standards for school meals, while expanding wellness policy requirements [15]. Regulations issued in 2016 provided guidance on local school wellness policy [7], requiring that foods sold outside of school meals and marketed on school campuses meet Smart Snacks in School standards [16].

Several evidence-informed practices have been recommended by entities, such as the USDA, to promote student participation in school meals programs [12,17]. One practice to enhance student health behaviors is the engagement of students themselves [18,19,20]. For example, during a formative evaluation of the SWITCH^®^ school wellness intervention, Chen et al. [18] gathered student perceptions on the implementation of educational modules and environmental changes in classroom, physical education and lunchroom settings; schools that engaged students in programming and decision-making reported higher implementation and satisfaction with the program. Gosliner (2014) found that integrating students in food-related decisions in the cafeteria (i.e., menu items) was positively associated with subsequent vegetable consumption at lunchtime, indicating that gaining student engagement and interest may influence behaviors. Jomaa et al. [20] found a significant association between comprehensiveness and rigor of district wellness policies, as measured through the Wellness School Assessment Tool (WellSAT) [21], and the inclusion of student-involvement goals in policies, demonstrating the importance of a comprehensive approach to the promotion of student health behaviors. Collectively, these findings demonstrate the importance of student engagement in fostering improved health behaviors. To date, however, few studies have considered whether policies at the district and state level that address student engagement impact school practices.

In addition to student engagement, the engagement of parents is important. A comprehensive intervention in Australia engaged schools and families in collaborative wellness programming to enhance nutrition and physical activity behaviors; parents in intervention schools reported increased fruit and vegetable consumption versus control schools [22]. The key strategies included sending families newsletters, co-creating healthy lunch resources by involving schools and families, and conducting nutrition education with parents. These findings are supported by the literature emphasizing the importance of simple and feasible parent engagement strategies [23,24]. Involving school nutrition staff in wellness programming and policy development is recommended [1,25], but little is known about the level of involvement typically occurring in school settings. Moreover, engaging nutrition managers and staff as key stakeholders facilitates the implementation of wellness policies [18,26,27], but little inquiry has been conducted into how district and state policies relate to such school-level practices.

The purpose of this study was to investigate the degree to which district and state-level policies relate to school practices, and to investigate differences in meal promotion practices via school demographic characteristics.

## 2. Materials and Methods

This study utilized school-level data from the School Nutrition and Meal Cost Study (SNMCS), conducted in the 2014–2015 school year, linked to objectively compiled data on state laws and district policies from the National Wellness Policy Study [8,9]. A de-identified linked data set was provided to the University of Illinois Chicago by Mathematica Policy Research, for analysis. This study was deemed to “not involve human subjects” by the University of Illinois Chicago Institutional Review Board (protocol #2020-0448).

### 2.1. Data and Design

A total of 1282 schools within 404 school food authorities in the 48 contiguous states and the District of Columbia (DC) participated in SNMCS, and 1210 completed the School Nutrition Manager Survey (96.9% weighted response rate) which was used for these analyses. School-level data from SNMCS are nationally representative of the public, non-charter schools that participate in the National School Lunch Program. Demographic information for the schools in these analyses is provided below in Table 1. Statistics in this paper may differ from those in the SNMCS Report [28] due to the different analytical sample used herein.

Detailed methods for SNMCS are provided elsewhere [29]. School food authorities were recruited via mailed materials with telephone follow-up, after which sampled schools within the school food authorities were recruited. The School Nutrition Manager Survey was completed online by school nutrition managers and included questions regarding kitchen characteristics, meal scheduling and pricing, implementation of meal and competitive food standards, and nutrition-promotion activities. This study used two questions: (1) “Does your school routinely make information on the calorie or nutrient content of USDA-reimbursable meals available to students or parents?” (response options “yes” or “no”); and (2) “Have you or anyone on your staff engaged in the following activities? Which have been adopted since SY 2012–2013 when the new meal patterns and nutrient standards for reimbursable lunches went into effect?” (response options included 19 specific activities and an “other” option). This study examined seven activities from the latter question: (1) involved students in planning school meal menus; (2) sought student input into vegetable offerings in school meals; (3) sought student input into creative or descriptive names for school meal dry bean and pea entrée items; (4) encouraged children to select fruit; (5) conducted a taste test activity with students; (6) set up a booth at a school event to promote or inform about school meals (for example, a family night or parent-teacher conference night); and (7) participated in a school or district meeting about the local wellness policy. Response options included “no,” “yes, since before SY 2012–2013,” and “yes, new since SY 2012–2013.” In addition to analyses of individual items, a school nutrition promotion scale was created based on items 1 to 5 to indicate how many strategies each school used to engage students (total scores from 0 to 5).

### 2.2. Control Measures

School-level variables were also obtained from SNMCS including: grade level (elementary, middle or high school), student racial/ethnic distribution (categorized as ≥66% white, ≥50% black, ≥50% Hispanic, and other), percentage of students eligible for free/reduced-price lunch (categorized by tertiles for the overall SNMCS school sample as ≤37.42%, >37.42–63.37% and >63.37%), urbanicity (urban, suburban and rural), size (<500, 500–999 and ≥1000 students), and Census region (West, Midwest, South and Northeast) [30]. Demographic data were included with the SNMCS dataset and were derived from data from the School Food Authority Director Survey and the National Center for Education Statistics’ Common Core of Data 2011–2012 [31]. Among the 1210 schools for which the School Nutrition Manager Survey was completed, 76 were missing responses to all questions of interest and another 9 were missing data on control variables, leaving 1125 schools. Additional item-specific missing data on survey questions and missing district policy data left 1056−1106 schools for analysis.

### 2.3. Policy Measures

Detailed methods for state law and district policy collection and coding are provided elsewhere [8,9]. State statutes and administrative regulations (collectively referred to as laws) were collected for all 50 states and DC from commercial legal databases Lexis Advance [32] and WestlawNext [33]. District policies were collected through internet searches, and calls and emails when necessary. District policies and state laws were deemed relevant if in effect as of the start of the 2014–2015 school year. All district policies and state laws were scored using a coding scheme developed by the National Wellness Policy Study [34], separately for each grade level (elementary, middle and high school). For this study, five policy variables were utilized; with one exception noted below, all were coded using an ordinal coding scheme with 0 equating to “no policy/provision,” 1 equal to a “weak” (encouraged or suggested) policy, and 2 equal to a “strong” (definitively required) policy. Where details of the coding vary, we specify it in the descriptions that follow. For the district policy data, one variable, SM11 (Nutrition Information for School Meals), was coded accordingly: a 1 refers to whether nutritional information related to calories, fats, sugars, etc., was only available upon request or was encouraged or suggested; a level 2 coding included specific requirements regarding nutrition information.

For state laws, four variables were considered. The first, SM4 (Strategies to Increase Participation), uses the no/weak/strong coding noted above, and refers to policy provisions focused on increasing student participation in school meals, such as through use of grab-and-go programs, student input on menus, taste tests and altered bus schedules. The remaining three state law variables used in this analysis relate to stakeholder involvement and feedback. The first variable, CP1 (Stakeholder Involvement in Development of Wellness Policy), measures whether state laws require that districts or district wellness policies address the extent to which the eight stakeholders required to be involved in wellness policy development by the Healthy, Hunger-Free Kids Act of 2010 are involved; among the required stakeholders are students and parents. For the purposes of this analysis, a level 1 coding equates to state law requiring stakeholders, but not all eight of the required stakeholders, a level 2 coding requires the six stakeholders (including students and parents) originally required to be involved in wellness policy development, and a level 3 coding involves all eight stakeholders (including students and parents) required as part of the Healthy, Hunger Free Kids Act. For this analysis, no state had level 3 coding, so a “strong” state law corresponded to level 2. Second, CP2 (Methods to Solicit Input) captured state laws that encouraged (level 1 coding) or required (level 2 coding) districts or district policies to obtain input from stakeholders including students (e.g., taste tests, food selection, menus). Finally, CP3 (How to Engage Parents/Community Members) examined whether state law encouraged (level 1 coding) or required (level 2 coding) districts or district policies to provide nutrition education (including nutrient information as well as menus, etc.) to parents/community members through newsletters, school events and other avenues.

### 2.4. Data Analysis

Logistic regressions were computed linking each survey item to corresponding district policy or state law provisions, controlling for school-level characteristics. For two of the seven nutrition promotion strategies, regressions were not conducted: seeking student input into creative or descriptive names for school meal dry bean and pea entrée items was relatively uncommon (20.6% of schools), and encouraging children to select fruit was nearly universal (96.4% of schools). The five-item nutrition promotion index had acceptable reliability (KR-20 = 0.67) [35], with a mean of 2.84 (SD = 1.41). Poisson regression was used to assess the association between the numbers of nutrition promotion strategies and state laws on strategies to increase participation in school meals. Adjusted prevalence estimates (after logistic regressions) or adjusted means (after Poisson regression) were computed, which allow comparisons of the outcomes at different levels of policy/law. Analyses were conducted in Stata/SE 15.1, considering the survey design and weights.

## 3. Results

As seen in Table 1, schools were from all regions of the United States, with a range of sizes (i.e., enrollment), populations (i.e., race, socio-economic status [SES]) and locale.

Table 2 shows descriptive statistics for survey data and policy/law. Nearly all schools reported actively encouraging fruit consumption (96%), and a majority reported taste tests (69%), school nutrition manager participation in school/district-level wellness policy-related meetings (69%), providing nutrient/calorie information for school meals (63%), and seeking student input on vegetable offerings in meals (53%).

Nearly 25% of schools were in a district with a policy that addressed nutrition information for school meals. Over one quarter of schools were in a state with a law regarding student participation in meals—23% of schools were in a state that encouraged and nearly 3% were in a state that required strategies to increase participation in school meals. Most schools (92%) were in a state that did not include provisions encouraging or requiring districts to involve key stakeholders in wellness policy development, and approximately three quarters of schools were in a state that did not encourage or require districts to obtain input from stakeholder groups, including students, or engage parents/community members; for the latter, when addressed, it tended to be required (23%).

Results of logistic regressions are shown in Table 3. Considering the association between district policy on nutrition information for school meals and schools routinely providing calorie/nutrient information on school meals, there was a significantly higher prevalence of this practice at schools with strong (required) district policies, as compared to no policy (AOR = 2.59, 95% CI 1.33, 5.05); the adjusted prevalence of schools providing calorie/nutrition information for meals was nearly 79% for schools in districts with a strong policy, versus 61% with no district policy. In addition to district policy, several school demographic characteristics were associated with this practice, with a higher likelihood of providing calorie/nutrient information in schools with majority Hispanic students (AOR = 2.51, 95% CI 1.02, 6.20) and those with mixed race/ethnicity (AOR = 1.97, 95% CI 1.06, 3.67), as compared to schools with at least 66% non-Hispanic white students. This practice was also more common at schools in the South versus the West (AOR = 2.60, 95% CI = 1.28, 5.28), and less common in rural than urban schools (AOR = 0.35, 95% CI 0.17, 0.71).

Table 4 shows the results from logistic regressions, exploring factors associated with school nutrition promotion activities, considering state law as a predictor. Results show a consistent association between state laws and school-level strategies. First, for involving students in planning menus, and seeking student input on vegetables, the prevalence of state laws addressing strategies to increase student participation in school meals was very low for strong laws (only 2.74% of schools), so this variable was collapsed as “any policy” (strong or weak) versus no policy. As shown in Table 4, involving students in menu planning was significantly more common where there was any state law (adjusted prevalence: 58.3% of schools) versus no law (adjusted prevalence: 41.3% of schools; AOR = 2.02, 95% CI 1.24, 3.31). Likewise, the practice of seeking student input on vegetables was more common where there was any state law (adjusted prevalence: 65.2% of schools) versus no law (adjusted prevalence: 48.8% of schools; AOR = 2.00, 95% CI 1.23, 3.24). Analyses examining the association between state law and the conducting of taste tests utilized the policy variable of methods for soliciting student input. Schools in states with weak (encouraged) policies were more likely to have conducted taste tests than those in states with no policy (adjusted prevalence of 80.2% vs. 67.7%; AOR = 2.02, 95% CI 1.06, 3.86), but having a strong (required) law did not significantly differ from no law.

With regard to the two practices to engage other stakeholders, such as families, there were significant associations between state law and these two practices. Schools were significantly more likely to use the strategy of setting up booths or displays at school events to advertise school meals when they were located in states with a strong (required) law on how to engage parents/community members, versus those in a state with no law (adjusted prevalence 47.9% vs. 33.2%; AOR = 1.93, 95% CI 1.07, 3.46). School nutrition manager/staff participation in wellness policy meetings was significantly more likely among schools in states with strong (required) or weak (encouraged) laws on involving stakeholders in wellness policy development, versus those in states with no laws, with an adjusted prevalence of 91.8% using this strategy in states with a strong law (AOR = 5.54, 95% CI 1.14, 26.88), and 87.4% in states with a weak law (AOR = 3.41, 95% CI 1.07, 10.85), compared to 67.6% with no law. Having school nutrition staff set up booths at parent–teacher conferences was less common in rural compared to urban schools (AOR = 0.35, 95% CI 0.19, 0.64), and schools with fewer than 500 students compared to those with at least 1000 students (AOR = 0.48, 95% CI 0.29, 0.80). Small schools were also less likely to conduct taste test activities with students (AOR = 0.38, 95% CI 0.22, 0.65).

Finally, a Poisson regression model was calculated to examine whether state law on strategies to increase participation in school meals and school characteristics were associated with the number of strategies used at each school to increase student participation in such school meals (Table 5). Schools in states with laws addressing strategies to increase participation in school meals reported having 18% more of these nutrition-oriented practices than schools in states without any laws on this topic.

## 4. Discussion

This study examined the association between district- and state-level nutrition promotion laws/policies and associated school practices, reported by school nutrition managers. Certain practices, such as providing nutrition information on school meals, promoting student vegetable intake, engaging students and parents in wellness policy and menu development, and promoting involvement in school meals, were positively associated with district or state laws and policies. Results are consistent with prior literature and provide compelling evidence for comprehensive state and district-level policies to enhance school food environments [4,13].

The most commonly used nutrition promotion strategies were related to promoting healthy food choices through encouraging students to select fruit, as well as conducting taste tests, which both engage students in decision-making processes. These collaborative strategies have been found to be successful in fostering healthy choices in the lunchroom [36,37], thus providing a feasible approach to promoting high-quality nutritional consumption. Less commonly used strategies were related to asking for student input on menu planning and specific menu items. This may be due to the time required to implement these strategies, in that nutrition managers may not be able to meet with students directly to plan menus and seek explicit feedback, as lack of time has been found to impede nutritional managers’ involvement in school wellness initiatives and nutrition policy implementation [38,39]. In addition, the level of skill and knowledge required to plan menus in accordance with USDA requirements may limit opportunities to involve students, particularly at the elementary and middle school level, making it hard for food service directors to implement this strategy.

Stronger district policies pertaining to providing nutrition information on school meals were related to corresponding actions, reflecting prior research on state nutrition policy strength [40]. However, the multivariable results also indicated that even after accounting for policy characteristics, there were persistent patterns of variation by school demographics. Specifically, rural schools were less likely than urban schools to provide nutrition information on USDA-reimbursable school meals and advertise school meal programs to parents at school events. Given the unique challenges of rural schools, such as access to fresh foods and variation in menu offerings [41], it may be more difficult to implement nutrition promotion strategies.

Several promising findings demonstrate the association between state law and school practices to promote student engagement in nutrition-related activities. Engaging students in programmatic decision making is an important strategy for schools [42,43,44], and having a state law that addressed strategies for increasing student participation in meals was a key predictor of whether schools engaged students in specific ways, such as seeking input on menu items and vegetable offerings. Comprehensive approaches to promoting nutrition through environmental changes and interactive programming in the lunchroom have yielded moderate success [45,46], yet the policy context surrounding these approaches remains poorly understood. The current results showing an association between policy and environmental/lunchroom practices, such as conducting taste test activities with students, suggest that policy may be one way to facilitate the implementation of such practices. This finding also underscores the need for language in policies/laws pertaining to school nutrition program participation. However, it is clear that policy alone is insufficient to facilitate environmental/strategy changes in schools. Understanding barriers and facilitators can be valuable for considering implementation support strategies. Qualitative research demonstrates the need for professional development and technical support to help nutrition managers implement policy [39].

Finally, the finding that the number of nutrition promotion strategies used in each school was positively related to the presence of state law concerning strategies to increase participation in school meals further supports the role that states can play in supporting school practices. Given that only 25% of the sample were in states that had any such policy, there is much room for improvement. Research with California State Board of Education representatives highlighted barriers to policy implementation, such as “competing interests” (e.g., core subjects) and lack of evidence about the impact of nutrition policy on academic achievement [47]. State lawmakers perceive that school obesity prevention policy has largely been ineffective [48]. To mitigate such barriers, it may be necessary to intentionally disseminate information to lawmakers through policy briefs, publishing in policy journals, and educating policymakers about the value of evidence-informed strategies in schools [49,50,51,52].

Despite the promising findings, there are several limitations. First, this was a cross-sectional study, which limits our ability to assess causality or to detect whether school practices changed over time. Follow-up research is necessary in order to investigate whether changes in national-, state- and district-level policy affect school practices. While the coding of policies/laws was objective and based on principles of legal epidemiology, the information provided by school nutrition managers was based on self-report, and may be subject to inaccurate knowledge or response biases. Lastly, more needs to be known about strategies that influence the successful implementation of policies. For example, it will be important to understand mechanisms of change, such as how contextual school-level or community-level characteristics (e.g., locale, capacity, funding) might moderate the relationship between policy and the implementation of nutrition-promotion practices in schools. Further research grounded in implementation science [53] is warranted.

## 5. Conclusions

This study provides information underscoring the rationale for developing and implementing strong policies at the state and district level in order to promote nutrition in schools. The low prevalence of such policies indicates an opportunity for greater advocacy, in order to engage lawmakers, and to encourage school districts to continue to review, monitor and strengthen their local school wellness policies. Identifying ways in which policies are implemented, providing context-specific guidance, and studying the impact of implementation on health outcomes will mark a significant step in bridging the gap between research and practice.

## Figures and Tables

**Table 1 nutrients-12-02356-t001:** Survey-weighted school sample characteristics.

Variable	Weighted Percent (%)
Grade Levels Offered	
Elementary	60.43
Middle	17.46
High	22.11
School Race	
≥66% White	53.05
≥50% Black	9.23
≥50% Hispanic	13.76
Other	23.96
Tertiles of % students eligible for FRPL	
Low (≤37.42%)	27.96
Medium (>37.42–63.37%)	32.61
High (>63.37%)	39.43
Locale	
Urban	22.36
Suburban	42.58
Rural	35.06
School Size	
Small (fewer than 500 students)	48.68
Medium (500 to 999 students)	39.77
Large (1000 or more students)	11.55
Census Region	
West	21.05
Midwest	28.25
South	35.80
Northeast	14.90

Note: *n* = 1125 schools; FRPL: free/reduced-price lunch.

**Table 2 nutrients-12-02356-t002:** Survey-weighted prevalence of school nutrition promotion-related practices and related district policy and state laws.

Variable	Weighted Percent (%) ^1^
School nutrition promotion practices	
Encouraged children to select fruit	96.41
Conducted a taste test activity with students	69.46
School nutrition manager (or staff) participation in school/district meeting on local wellness policy	69.31
School routinely provides calorie/nutrient information on USDA-reimbursable meals	63.35
Sought student input into vegetable offerings in school meals	53.06
Involved students in planning school meal menus	45.52
Set up booth at school event to advertise school meals	36.33
Sought student input on names for dry bean/pea entrée items	20.55
District policy on nutrition information for school meals	
No policy	75.10
Weak policy	12.14
Strong policy	12.76
State law on strategies to increase participation in school meals	
No policy	74.52
Weak policy	22.74
Strong policy	2.74
State law on stakeholders involved in wellness policy development	
No policy	92.00
Weak policy	5.31
Strong policy	2.68
State law on methods to solicit input	
No policy	76.07
Weak policy	14.99
Strong policy	8.94
State law on how to engage parents/community	
No policy	74.99
Weak policy	1.87
Strong policy	23.14

Note: *n* = 1081–1125 schools due to item-specific missing data; ^1^ The weighted percent of district policy and state laws represent the weighted percent of schools included in the analysis that were located in a district or state, respectively, with the given policy or law on the books.

**Table 3 nutrients-12-02356-t003:** Logistic Regression Results for the Association between District Policy on Nutrition Information for School Meals and Schools Routinely Providing Calorie/Nutrient Information for USDA-Reimbursable Meals.

Variable	AOR (95% CI)
District policy on nutrition information for school meals	
No policy/provision (Ref)	1.00 (1.00, 1.00)
Weak policy	2.22 (0.97, 5.07)
Strong policy	2.59 ** (1.33, 5.05)
Grade Levels Offered	
Elementary (Ref)	1.00 (1.00, 1.00)
Middle	0.95 (0.69, 1.30)
High	0.76 (0.52, 1.10)
School Race	
≥66% White (Ref)	1.00 (1.00, 1.00)
≥50% Black	1.33 (0.53, 3.31)
≥50% Hispanic	2.51 * (1.02, 6.20)
Other	1.97 * (1.06, 3.67)
Tertiles of % students eligible for FRPL	
Low (≤37.42%) (Ref)	1.00 (1.00, 1.00)
Medium (>37.42–63.37%)	0.73 (0.45, 1.19)
High (>63.37%)	0.57 (0.27, 1.22)
Urbanicity	
Urban (Ref)	1.00 (1.00, 1.00)
Suburban	0.69 (0.37, 1.30)
Rural	0.35 ** (0.17, 0.71)
School Size	
Small (fewer than 500 students)	0.70 (0.41, 1.19)
Medium (500 to 999 students)	0.77 (0.47, 1.24)
Large (1000 or more students) (Ref)	1.00 (1.00, 1.00)
Census Region	
West (Ref)	1.00 (1.00, 1.00)
Midwest	1.73 (0.82, 3.65)
South	2.60 ** (1.28, 5.28)
Northeast	1.08 (0.49, 2.35)
N of schools in analysis	1056
Adjusted prevalence of schools providing calorie/nutrition information for meals by district policy strength	
No district policy on nutrition information for school meals	60.5%
Weak district policy on nutrition information for school meals	76.0%
Strong district policy on nutrition information for school meals	78.6%

* *p* < 0.05, ** *p* < 0.01; AOR: Adjusted odds ratio; CI: confidence interval; Ref: referent category; FRPL: free/reduced-price lunch.

**Table 4 nutrients-12-02356-t004:** Logistic Regression Results for the Association between State Law and School Nutrition Manager Activities.

Variable	Involved Students in Planning Menus ^1^	Sought Student Input on Vegetables ^1^	Conducted A Taste Test Activity With Students ^2^	Set Up Booth at School Event to Advertise Meals ^3^	Participated in Meeting on Wellness Policy ^4^
	AOR (95% CI)	AOR (95% CI)	AOR (95% CI)	AOR (95% CI)	AOR (95% CI)
State Law Categorization					
No policy (Ref)	1.00 (1.00, 1.00)	1.00 (1.00, 1.00)			
Any policy	2.02 ** (1.24, 3.31)	2.00 ** (1.23, 3.24)			
No policy (Ref)			1.00 (1.00, 1.00)	1.00 (1.00, 1.00)	1.00 (1.00, 1.00)
Weak policy			2.02 * (1.06, 3.86)	0.70 (0.15, 3.22)	3.41 * (1.07, 10.85)
Strong policy			0.96 (0.45, 2.02)	1.93 * (1.07, 3.46)	5.54 * (1.14, 26.88)
Grade Levels Offered					
Elementary (Ref)	1.00 (1.00, 1.00)	1.00 (1.00, 1.00)	1.00 (1.00, 1.00)	1.00 (1.00, 1.00)	1.00 (1.00, 1.00)
Middle	0.97 (0.71, 1.31)	1.10 (0.78, 1.54)	0.85 (0.57, 1.25)	1.00 (0.71, 1.42)	1.16 (0.83, 1.62)
High	1.00 (0.70, 1.44)	1.22 (0.84, 1.77)	0.80 (0.53, 1.19)	0.77 (0.52, 1.16)	1.25 (0.84, 1.84)
School Race					
≥66% White (Ref)	1.00 (1.00, 1.00)	1.00 (1.00, 1.00)	1.00 (1.00, 1.00)	1.00 (1.00, 1.00)	1.00 (1.00, 1.00)
≥50% Black	1.47 (0.61, 3.55)	1.20 (0.51, 2.80)	0.86 (0.35, 2.14)	0.53 (0.21, 1.34)	1.66 (0.66, 4.20)
≥50% Hispanic	1.08 (0.52, 2.26)	0.78 (0.37, 1.63)	0.72 (0.34, 1.52)	0.75 (0.33, 1.69)	1.93 (0.88, 4.25)
Other	1.23 (0.73, 2.06)	1.08 (0.68, 1.74)	1.19 (0.67, 2.13)	0.69 (0.42, 1.13)	1.36 (0.81, 2.30)
Tertiles of % students eligible for FRPL					
Low (≤37.42%) (Ref)	1.00 (1.00, 1.00)	1.00 (1.00, 1.00)	1.00 (1.00, 1.00)	1.00 (1.00, 1.00)	1.00 (1.00, 1.00)
Medium (>37.42–63.37%)	1.09 (0.70, 1.70)	1.04 (0.65, 1.64)	1.00 (0.61, 1.66)	0.91 (0.58, 1.41)	0.83 (0.50, 1.38)
High (>63.37%)	0.75 (0.43, 1.32)	1.25 (0.70, 2.25)	0.90 (0.49, 1.68)	1.37 (0.74, 2.52)	1.12 (0.59, 2.15)
Urbanicity					
Urban (Ref)	1.00 (1.00, 1.00)	1.00 (1.00, 1.00)	1.00 (1.00, 1.00)	1.00 (1.00, 1.00)	1.00 (1.00, 1.00)
Suburban	1.31 (0.76, 2.24)	0.98 (0.58, 1.64)	0.87 (0.49, 1.55)	0.67 (0.40, 1.13)	1.02 (0.58, 1.79)
Rural	1.40 (0.76, 2.58)	1.36 (0.77, 2.41)	0.59 (0.31, 1.11)	0.35 *** (0.19, 0.64)	1.39 (0.74, 2.61)
School Size					
<500 students	1.02 (0.61, 1.70)	1.12 (0.66, 1.91)	0.38 *** (0.22, 0.65)	0.48 ** (0.29, 0.80)	1.07 (0.63, 1.80)
500–999 students	1.01 (0.63, 1.62)	0.92 (0.57, 1.49)	0.91 (0.54, 1.56)	0.60 * (0.38, 0.96)	0.89 (0.54, 1.45)
≥1000 students (Ref)	1.00 (1.00, 1.00)	1.00 (1.00, 1.00)	1.00 (1.00, 1.00)	1.00 (1.00, 1.00)	1.00 (1.00, 1.00)
Census Region					
West (Ref)	1.00 (1.00, 1.00)	1.00 (1.00, 1.00)	1.00 (1.00, 1.00)	1.00 (1.00, 1.00)	1.00 (1.00, 1.00)
Midwest	1.58 (0.85, 2.93)	1.05 (0.54, 2.04)	1.10 (0.55, 2.20)	1.38 (0.73, 2.62)	1.58 (0.82, 3.06)
South	1.55 (0.87, 2.78)	0.76 (0.42, 1.38)	0.74 (0.39, 1.39)	0.73 (0.36, 1.47)	1.11 (0.60, 2.08)
Northeast	2.16 * (1.08, 4.29)	1.25 (0.61,2.60)	1.33 (0.59, 3.01)	1.21 (0.55, 2.65)	2.00 (0.90, 4.43)
N of schools in given analysis	1105	1106	1104	1101	1099
Adjusted prevalence of school nutrition manager activity based on state law strength					
No policy	41.3%	48.8%	67.7%	33.2%	67.6%
Any policy/Weak policy	58.3%	65.2%	80.2%	26.3%	87.4%
Strong policy			66.8%	47.9%	91.8%

^1^ School nutrition manager activities related to involving students in planning school meal menus and seeking student input on vegetable offerings in school meals were linked to state law on strategies to increase participation in school meals; ^2^ Conducting a taste test activity with students was linked to state law on methods to solicit input; ^3^ Setting up a booth at a school event to advertise school meals was linked to state law on how to engage parents/community; ^4^ Participating in a school or district meeting about the local wellness policy was linked to state law on stakeholders involved in wellness policy development; * *p* < 0.05, ** *p* < 0.01, *** *p* < 0.001; AOR: Adjusted odds ratio; CI: confidence interval; Ref: referent category; FRPL: free/reduced-price lunch.

**Table 5 nutrients-12-02356-t005:** Poisson Regression Results for the Association Between State Law on Strategies to Increase Participation in School Meals and School Nutrition Manager Activities Index.

Variable	IRR (95% CI)
State Law Categorization	
No law on strategies to increase participation in meals (Ref)	1.00 (1.00, 1.00)
Any law on strategies to increase participation in meals	1.18 ** (1.05, 1.32)
Grade Levels Offered	
Elementary (Ref)	1.00 (1.00, 1.00)
Middle	1.00 (0.92, 1.08)
High	1.00 (0.92, 1.09)
School Race	
≥66% White (Ref)	1.00 (1.00, 1.00)
≥50% Black	1.12 (0.88, 1.42)
≥50% Hispanic	1.01 (0.84, 1.21)
Other	1.05 (0.93, 1.18)
Tertiles of % students eligible for FRPL	
Low (≤37.42%) (Ref)	1.00 (1.00, 1.00)
Medium (>37.42–63.37%)	0.98 (0.88, 1.09)
High (>63.37%)	0.93 (0.81, 1.06)
Urbanicity	
Urban (Ref)	1.00 (1.00, 1.00)
Suburban	1.01 (0.89, 1.15)
Rural	0.98 (0.84, 1.14)
School Size	
Small (fewer than 500 students)	0.94 (0.84, 1.06)
Medium (500 to 999 students)	1.00 (0.91, 1.11)
Large (1000 or more students) (Ref)	1.00 (1.00, 1.00)
Census Region	
West (Ref)	1.00 (1.00, 1.00)
Midwest	1.12 (0.96, 1.30)
South	1.07 (0.94, 1.23)
Northeast	1.15 (0.98, 1.35)
N of schools in analysis	1077
Adjusted mean score for school nutrition manager activities index (out of 5)	
Without state law on strategies to increase participation in meals	2.7
With state law on strategies to increase participation in meals	3.2

** *p* < 0.01; IRR: Incidence rate ratio; CI: confidence interval; Ref: referent category; FRPL: free/reduced-price lunch.

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
