# Peer review of "Assessing the Relationship between District and State Policies and School Nutrition Promotion-Related Practices in the United States"

_nutrients, 2020, doi:10.3390/nu12082356_

Round 1
Reviewer 1 Report
good choice of interesting topic. The results show what would be expected re school environment and state and strict laws. The stronger the regulation, the bigger effect.
The assessment of the role of state and district policies on school environment ibrings a good case for insisting on development of these policies in the states and districts where they do not exist. The research confirms that those policies have positive impact on school environment and engagement of students.
Methodology used is adequate, and the results and discussion are clearly presented.
Author Response
Nutrients-857032 Response to Reviewers
Reviewer 1:
good choice of interesting topic. The results show what would be expected re school environment and state and strict laws. The stronger the regulation, the bigger effect.
The assessment of the role of state and district policies on school environment brings a good case for insisting on development of these policies in the states and districts where they do not exist. The research confirms that those policies have positive impact on school environment and engagement of students.
Methodology used is adequate, and the results and discussion are clearly presented.
Thank you for this expeditious review; we greatly appreciate your feedback and interest in this paper.
Reviewer 2 Report
This paper evaluates the link between healthy eating policies and school nutrition practices in a larger number of schools throughout the United States. The results show that a number of nutrition practices are more common among schools that reside in districts that have such policies.
This is a nice paper. It has a large and diverse sample of schools, evaluates relevant nutrition practices, it appears to be conducted well, and the conclusions of the paper follow logically from the results. The results of this paper should be of value to the reader’s of this journal.
The main limitation of the paper is that the outcome variables are self-report (teacher reported?) data rather than objective data. Other than this issue, my comments mostly concern the presentation/write up of the paper:
- The introduction is clear and provides a good rationale for the research.
Minor comments for readability:
The sentence on line 63-64, suggest changing “addressing” to “that address”.
Line 74: “manifests organically”. I don’t know what the authors are trying to say here. Could this be replaced with a more concrete/precise phrasing?
- The authors use too many acronyms which makes the paper hard to read. I would strongly recommend writing the following acronyms out in full every time: UIC, SFAs, SNMCS, HHFKA. UIC isn’t written again, so I’m not sure why adding the acronym is necessary. Same with CNR. More commonly used acronyms (such as DC and SES are fine).
- In the methods section (line 87-111) the authors describe the data collected from schools however it is not clear who provided the information from each school. It’s called the school nutrition manager survey – does that mean it was a nutrition manager? Do all schools in your sample have these managers?
- The key limitation of this study is the reliance on staff-reported data to assess the school nutrition practices. The sentence in the abstract on line 15-17 should be changed to:
“This study examined how state and district policies are associated with *teacher-reported* school practices to promote student participation in school lunch programs”.
Or similar wording depending on the answer to point 3. The authors do address this clearly in the discussion section and therefore no further change is needed there. The authors do use the term “self-report” however – is this accurate? Given that a school cannot self-report? Perhaps better to describe as subjective assessment of the school’s practices from the point of view of a single teacher/staff/etc from that school, or a similar wording.
- Line 111: “school level controls…”. Recommend using the more precise word variables instead of controls.
- I recommend re-structuring the methods section with subtitles. Could include: Schools, Measures(or outcomes), Policies, etc
- Throughout the document the authors refer to “wellness”. While in some cases this appears to be an official name of a policy – the authors also use the term more generally. For example “school wellness environments” p12, line 18. There are existing frameworks for talking about environments (eg Swinburn et al’s 1999 Angelo framework: physical environments, sociocultural environments). Food environments is also used often. I’m not familiar with “wellness environments” being used regularly. This seems vague and I’m not sure what this actually means. I would recommend changing the wording to healthy eating policies/food environments. Can the authors justify using the term wellness?
- P3, line 131-133. This isn’t entirely clear, are the authors saying that a “weak” coding means that the nutrition practice is encouraged but not mandated? Whereas strong policies are mandated?
Author Response
Reviewer 2:
This paper evaluates the link between healthy eating policies and school nutrition practices in a larger number of schools throughout the United States. The results show that a number of nutrition practices are more common among schools that reside in districts that have such policies.
Thank you very much for taking the time to complete this review. We greatly appreciate the feedback you have provided and have taken time to address the comments presented below; all responses are in italic font and we have provided line numbers where relevant.
This is a nice paper. It has a large and diverse sample of schools, evaluates relevant nutrition practices, it appears to be conducted well, and the conclusions of the paper follow logically from the results. The results of this paper should be of value to the readers of this journal.
The main limitation of the paper is that the outcome variables are self-report (teacher reported?) data rather than objective data. Other than this issue, my comments mostly concern the presentation/write up of the paper:
- The introduction is clear and provides a good rationale for the research.
Minor comments for readability:
The sentence on line 63-64, suggest changing “addressing” to “that address”.
Thank you- this has been changed.
Line 74: “manifests organically”. I don’t know what the authors are trying to say here. Could this be replaced with a more concrete/precise phrasing?
We have revised this sentence to enhance readability and convey that there is relatively little information on how nutrition managers participate in wellness policies/practices in general and the relationship between district and state policies – please see lines 72-74.
- The authors use too many acronyms which makes the paper hard to read. I would strongly recommend writing the following acronyms out in full every time: UIC, SFAs, SNMCS, HHFKA. UIC isn’t written again, so I’m not sure why adding the acronym is necessary. Same with CNR. More commonly used acronyms (such as DC and SES are fine).
We have removed the following acronyms: UIC, SFA, and HHFKA. We did keep SNMCS since this is a long acronym and is the name of the overarching project, encompassing other papers in this special issue. We therefore wanted to remain consistent and ensure that acronyms were the same across other papers.
- In the methods section (line 87-111) the authors describe the data collected from schools however it is not clear who provided the information from each school. It’s called the school nutrition manager survey – does that mean it was a nutrition manager? Do all schools in your sample have these managers?
Yes- this assumption is correct. The surveys were completed by nutrition managers – we have clarified this on lines 99-100.
- The key limitation of this study is the reliance on staff-reported data to assess the school nutrition practices. The sentence in the abstract on line 15-17 should be changed to:
“This study examined how state and district policies are associated with *teacher-reported* school practices to promote student participation in school lunch programs”.
We have clarified that these practices were reported by school nutrition managers i.e.,“self-report”.
Or similar wording depending on the answer to point 3. The authors do address this clearly in the discussion section and therefore no further change is needed there. The authors do use the term “self-report” however – is this accurate? Given that a school cannot self-report? Perhaps better to describe as subjective assessment of the school’s practices from the point of view of a single teacher/staff/etc. from that school, or a similar wording.
- Line 111: “school level controls…”. Recommend using the more precise word variables instead of controls.
Thank you- we have made this change.
- I recommend re-structuring the methods section with subtitles. Could include: Schools, Measures(or outcomes), Policies, etc
Thank you for these suggestions- we have added the following headers in the methods section:
2.1 Data and Design
2.2. Control Variables
2.3 Data Analysis
- Throughout the document the authors refer to “wellness”. While in some cases this appears to be an official name of a policy – the authors also use the term more generally. For example “school wellness environments” p12, line 18. There are existing frameworks for talking about environments (eg Swinburn et al’s 1999 Angelo framework: physical environments, sociocultural environments). Food environments is also used often. I’m not familiar with “wellness environments” being used regularly. This seems vague and I’m not sure what this actually means. I would recommend changing the wording to healthy eating policies/food environments. Can the authors justify using the term wellness?
Thank you for this comment. The overarching term of “school wellness” was used in this article to conceptualize the multiple components of wellness policy implementation. In the SNMCS, and the questions pertaining to nutrition managers’ involvement in wellness policy, we did intend to capture global and nutrition-specific involvement. However, in some places in the article we have clarified this to pertain to nutrition policy/programming at the school level (i.e., P12, line 28).
- P3, line 131-133. This isn’t entirely clear, are the authors saying that a “weak” coding means that the nutrition practice is encouraged but not mandated? Whereas strong policies are mandated?
We apologize for this oversight. We have clarified the language on what is now lines 141-142 to read: “1 equal to a “weak” (encouraged or suggested) policy, and 2 equal to a “strong” (definitively required) policy….” (Highlighted language is only for emphasis for this response to the reviewer.)